# Postprandial response of leptin and adiponectin to standardized high-carbohydrate and high-fat meals in adults: A cross-sectional study

Fabiola Mabel Del Razo-Olvera[1,2☉], Ana Teresa Arias-Marroquín[1,3☉],
Mayra Martínez-Fajardo[2], Wendolyn Elideth Dávila-Olmedo[1], Ivette Cruz Bautista[1,3],
Gabriela Alejandra Galán-Ramírez[1,3], Liliana Juárez-Martínez[1],
Donají Verónica Gómez-Velasco[1], Luz Elizabeth Guillén-Pineda[4],
Angelina López Estrada[4], Rodrigo Otero-Otero[1], Daniel Elías-López[1,4],
Alexandro José Martagón-Rosado[1,3,5], Carlos Aguilar-Salinas[1,3], Adrian Soto-Mota[1,3*]

1 "Unidad de Investigación de Enfermedades Metabólicas", Instituto Nacional de Ciencias Médicas y Nutrición Salvador Zubirán, Mexico City, Mexico, 2 "Escuela de Altos Estudios en Salud", Universidad La Salle, Mexico City, Mexico, 3 Tecnologico de Monterrey, School of Medicine, Mexico City, Mexico, 4 Department of Endocrinology and Lipid Metabolism, Instituto Nacional de Ciencias Médicas y Nutrición Salvador Zubirán, Mexico City, Mexico, 5 OriGen Health Research Center, University of Texas at Austin & Tecnologico de Monterrey, Austin, Texas, United States of America

☉ These authors contributed equally to this work.
* adrian.sotom@incmnsz.mx

## Abstract

### Background

Leptin and adiponectin link adiposity with metabolic regulation. While their fasting levels have been extensively studied, little is known about their postprandial dynamics, and it is generally assumed that their concentration is unaffected by food intake.

### Objective

To evaluate the postprandial responses of leptin and adiponectin following two standardized meals differing in macronutrient composition, in adults with and without obesity.

### Methods

In 79 adults (43 women, 36 men; 47% with obesity), we investigated whether meal composition influences adipokine responses. Participants completed two standardized mixed-meal tolerance tests: a high-carbohydrate meal (60% carbohydrate, 20% fat, 20% protein) and a high-fat meal (50% fat, 30% carbohydrate, 20% protein). Blood samples were collected at baseline and after 60, 120, and 360 minutes. Data were evaluated using linear mixed-effects models.

**Data availability statement:** Data are part of Mexico's Biobank for Metabolic Diseases (BIOMEM). Due to ethical and legal restrictions related to participant privacy and the potentially sensitive nature of clinical and anthropometric data, the dataset cannot be made publicly available. Researchers who meet the criteria for access to confidential data may request access through the Biobanco Mexicano de Enfermedades Metabólicas (BIOMEM) at [biomem@incmnsz.mx](mailto:biomem@incmnsz.mx) or through the Institutional Research Ethics Committee of the Instituto Nacional de Ciencias Médicas y Nutrición Salvador Zubirán at [comite.etica.investigacion@incmnsz.mx](mailto:comite.etica.investigacion@incmnsz.mx). Requests will be reviewed according to institutional ethical and legal requirements for data access.

**Funding:** Mexican Society of Endocrinology (SMNE) and the Instituto Nacional de Ciencias Médicas y Nutrición Salvador Zubirán.

**Competing interests:** The authors have declared that no competing interests exist.

**Abbreviations:** MMTT, Mixed-Meal Tolerance Test; T2D, Type 2 Diabetes; BMI, Body Mass Index; CVD, Cardiovascular Disease; DXA, Dual-Energy X-ray Absorptiometry; VAT, Visceral Adipose Tissue; HDL-C, High-Density Lipoprotein Cholesterol.

## Results

Women exhibited higher leptin and adiponectin concentrations than men ($p < 0.001$). Participants with obesity had higher leptin and lower adiponectin across all time points ($p < 0.001$) with both meals. In mixed-effects models, Sex and BMI were significant predictors of both concentrations of adipokines. Meal type significantly influenced leptin trajectories, with leptin concentrations having a progressive decline after the high-fat meal; in contrast, they did not change significantly after the high-carbohydrate meal. Adiponectin concentration showed only modest time-related changes but was not significantly modified by meal composition.

## Conclusions

Sex and BMI are significant determinants of the postprandial dynamics of leptin and adiponectin, while meal composition, especially high-fat meals, influences leptin concentrations. These observations underscore the importance of considering postprandial dynamics rather than basal levels for assessing metabolic health. The reductions in leptin, particularly in individuals with obesity, could potentially influence satiety-related pathways, although this hypothesis requires confirmation in studies directly assessing appetite and energy intake.

## Introduction

Leptin and adiponectin are among the most extensively studied adipokines due to their roles in regulating energy balance, as well as glucose and lipid metabolism. Predominantly secreted by adipose tissue, leptin is also produced in smaller amounts by skeletal muscle, gastric mucosa, and salivary glands. [1]. It communicates with the central nervous system, particularly to the hypothalamus, regarding the body's energy reserves and, thereby, influences appetite and energy expenditure [2].

Although leptin is generally considered a long-term regulator of energy balance, evidence suggests that its secretion may also respond to dietary composition in the postprandial state. For example, resistant starch supplementation has been shown to reduce circulating leptin concentrations independently of changes in body composition [3]. In contrast, low-carbohydrate, high-fat diets have demonstrated variable effects, ranging from leptin resistance to reductions mediated by insulin and glucose dynamics [3,4].

Adiponectin exerts anti-inflammatory and insulin-sensitizing effects in receptors expressed in the liver, skeletal muscle, pancreas, and vasculature [5]. Low adiponectin levels are consistently associated with metabolic disorders such as type 2 diabetes (T2D), hypertension, and coronary artery disease [6–8].

Dietary factors, including monounsaturated and polyunsaturated fatty acids, fiber, and polyphenols, have been positively associated with adiponectin concentrations, whereas high-glycemic index diets tend to suppress its circulating levels [6,7]. These adipokines are regulated through complex mechanisms, often showing paradoxical

associations: elevated leptin levels (particularly, leptin resistance) are correlated with an increased cardiometabolic risk. In contrast, excessively high adiponectin levels are observed in people with chronic inflammation or advanced cardiovascular disease (CVD) [5].

Despite extensive research on fasting levels, little is known about their postprandial dynamics, particularly regarding meals with differing macronutrient compositions. Findings remain inconsistent: some studies report no changes in leptin postprandially, while others describe increases in normal-weight individuals and decreases in those with obesity. For adiponectin, discrepancies have also been observed, likely attributable to differences in study design, meal composition, and participant characteristics [6,9].

To address the gaps, we aimed to evaluate leptin and adiponectin trajectories during a mixed-meal tolerance test (MMTT) after two standardized meals differing in macronutrient composition, in adults with and without obesity. This design allowed us to assess whether body composition and the dietary fat-to-carbohydrate ratio differentially modulate postprandial adipokine responses. Understanding these dynamics may help identify early markers of metabolic dysfunction in populations with high obesity prevalence.

## Methods

### Study design and participants

This cross-sectional study is part of the Mexican Biobank for Metabolic Diseases Project (BIOMEM) and was conducted at the Unidad de Investigación de Enfermedades Metabólicas (UIEM) at the Instituto Nacional de Ciencias Médicas y Nutrición "Salvador Zubirán" (INCMNSZ). The institutional Ethics in Human Research Committee assessed and approved the protocol (reference number 3722), and all participants provided written and witnessed informed consent (by paper-based autograph signature). Recruitment started on July 1st, 2022, and finished on July 31st, 2025. This study was conducted in accordance with the STROBE guidelines for cross-sectional studies.

The enrolled participants were Mexican adults with a body mass index (BMI) between 18.5 and 34.9 kg/m². Participants were excluded if they had a prior diagnosis of type 1 or type 2 diabetes, were pregnant or lactating, or were currently using medications known to affect glucose or lipid metabolism (e.g., immunosuppressants, antibiotics, statins, fibrates, or metformin). Additional exclusion criteria included chronic liver disease or serum transaminase levels ≥ 3 times the upper limit of normal, history of celiac disease or short bowel syndrome, previous bariatric surgery, or consumption of dietary supplements such as protein powders, creatine, or omega-3 fatty acids.

### Clinical and biochemical data

Blood pressure and vital signs were assessed at baseline. Anthropometric measurements included body weight, height, and waist and hip circumferences. Body composition was evaluated by dual-energy X-ray absorptiometry (DXA; GE Lunar Prodigy Advance, Encore software version 16.10), providing estimates of total fat mass, lean mass, and visceral adipose tissue (VAT). Obesity was defined as a body mass index (BMI) ≥ 30 kg/m².

Fasting venous blood samples were collected at baseline (0 min) and at 60, 120, and 360 minutes after meal ingestion to measure leptin (ng/mL) and adiponectin (µg/mL) concentrations. Serum levels were measured using commercially available ELISA kits (Millipore®, Sigma-Aldrich, Burlington, MA, EZHL-80SK for leptin and EZHADP-61BK for adiponectin), following the manufacturer's protocols.

### Standardized test meals

Participants completed two test visits within the same week, each visit involving a standardized mixed meal differing in macronutrient composition: a high-carbohydrate, low-fat meal (60% carbohydrate, 20% fat, 20% protein) and a high-fat, low-carbohydrate meal (50% fat, 30% carbohydrate, 20% protein). The order of meal administration was randomly

assigned for each participant. The interval between visits did not exceed seven days, as the outcomes evaluated were acute postprandial responses; no formal washout period was implemented. This approach was intentionally selected to avoid changes in habitual diet or lifestyle behaviours during acute postprandial responses under comparable baseline conditions. Both meals were designed to resemble a standard breakfast of this population. To minimize variability, all participants consumed a standardized dinner the evening before each test day. This consisted of a commercially formulated, standardized beverage (237 ml) manufactured under controlled pharmaceutical-grade conditions with the following composition: (63% carbohydrate, 22% fat, 15%). The specific food items and quantities are detailed in S1 Table.

The primary outcomes were serum leptin and adiponectin concentrations measured at 0, 60, 120, and 360 minutes after meal ingestion. The main exposure was meal type (high-carbohydrate vs. high-fat standardized meal), and the time after meal ingestion was treated as a continuous predictor. Sex and BMI were included as key predictors. Obesity status (BMI ≥ 30 kg/m²) was evaluated as a potential effect modifier of postprandial adipokine responses via interaction terms with meal type. Potential confounders were addressed through strict eligibility criteria, standardized pre-test meals, and multivariable modeling.

Some strategies were implemented to minimize potential bias. We defined strict eligibility criteria, excluding individuals with diabetes, chronic liver disease, or medications known to affect glucose or lipid metabolism. Measurement bias was minimized by using standardized protocols for anthropometric assessments, DXA-derived body composition measurements, and commercially validated ELISA assays for adipokine quantification.

To reduce within- and between-subject variability, a repeated-measures design was employed in which all participants completed both meal challenges under standardized conditions. Potential confounding was addressed using multivariable mixed-effects models that adjusted for sex, BMI, time, and meal type.

## Statistical analysis

Using simr, we estimated that a sample size of n ~ 60 would yield a statistical power of at least 80% to observe physiologically relevant (defined as d = 0.6) differences in postprandial adiponectin and leptin between two groups with a linear mixed effects model that treats identity as a random effect and observes between 4 and 6 timepoints. However, more participants were recruited to allow for drop-outs and missing data.

All statistical analyses were performed using R version 4.4.3 (R Foundation for Statistical Computing) with RStudio version 2025.05.0. Continuous variables are presented as mean, standard deviation (SD), and categorical variables as frequencies and percentages. Baseline characteristics were compared between groups using the Wilcoxon rank-sum test for continuous variables and the chi-square test for categorical variables. In the linear mixed-effects models, time was modeled as a continuous variable (in minutes), allowing the model to estimate a single average rate of change in adipokine concentration per minute across the postprandial period. Model assumptions were evaluated through residual diagnostics, including visual inspection of histograms and standard Q-Q plots of the residuals, which supported the assumption of approximate normality.

For postprandial analyses, leptin and adiponectin concentrations were assessed at baseline (0 min) and at 60, 120, and 360 min after each standardized meal (carbohydrate-rich and fat-rich). Between-group comparisons (by meal type or obesity status) at individual time points were performed using the Wilcoxon rank-sum test to assess between-group differences at specific postprandial time points, providing pointwise comparisons.

To evaluate overall postprandial trajectories across all time points, linear mixed- effects models were fitted using the lmer function from the *lmerTest* package, with Satterthwaite's method to estimate degrees of freedom [10]. Fixed effects included time, sex, BMI, and meal type, while individual participants were included as random intercepts to account for repeated measurements within participants. Model assumptions were evaluated using residual diagnostics, including histograms and standard Q-Q plots. Explained variance was quantified using conditional and marginal $R^2$ derived from the de *MuMIn: r.squaredGLMM* function.

Models included interaction terms between BMI and meal type. Model estimates (β) represent the rate of change in hormone concentration (ng/mL for leptin and μg/mL for adiponectin) per minute or the difference in rate of change per minute. Fixed effects included time, sex, BMI, and meal type, and the interaction between BMI and meal type. Individual participants were included as random intercepts to account for repeated measurements within participants.

A two-sided p-value <0.05 was considered statistically significant. All plots were produced using ggplot2. Missing data were treated intrinsically with the choice of linear mixed effects models. No data were imputed since there were no missing outcome data.

## Results

### Baseline characteristics of participants

A total of 79 participants were included (43 women and 36 men). Women showed significantly higher leptin (41.90 ± 24.27 vs. 17.43 ± 12.86 ng/mL, p < 0.001) and adiponectin concentrations (12.29 ± 5.51 vs. 8.09 ± 2.76 μg/mL, p < 0.001) than men. As well as higher HDL concentration levels, compared with men (Table 1). All participants completed both phases.

In contrast, men exhibited higher glucose and triglyceride concentrations, as well as higher systolic and diastolic blood pressure values. Regarding anthropometric measurements, men had greater height and weight, whereas women had higher fat mass. Overall, 46.5% of women (n = 20) and 47.2% of men (n = 17) were classified as having obesity, while 53.5% of women (n = 23) and 52.8% of men (n = 19) were classified as not having obesity (Table 1).

**Table 1. Baseline characteristics of the participants stratified by Sex.**

| Variable | Women (n = 43) | Men (n = 36) | *p*-value |
|---|---|---|---|
| **Biochemical parameters** | | | |
| Leptin (ng/mL) | 41.90 (24.27) | 17.43 (12.86) | <0.001 |
| Adiponectin (μg/mL) | 12.29 (5.51) | 8.09 (2.76) | <0.001 |
| Insulin (μU/mL) | 12.13 (7.11) | 15.23 (9.12) | 0.094 |
| Glucose (mg/dL) | 94.88 (13.85) | 102.58 (11.77) | 0.010 |
| Triglycerides (mg/dL) | 140.91 (64.47) | 182.47 (91.60) | 0.021 |
| Total cholesterol (mg/dL) | 185.86 (35.55) | 191.69 (38.21) | 0.485 |
| HDL cholesterol (mg/dL) | 49.93 (12.93) | 37.25 (7.84) | <0.001 |
| LDL cholesterol (mg/dL) | 108.37 (30.96) | 117.82 (30.75) | 0.216 |
| **Anthropometric measurements** | | | |
| Age (years) | 49.17 (15.96) | 52.49 (14.37) | 0.345 |
| Weight (kg) | 72.49 (16.07) | 89.27 (18.80) | <0.001 |
| Height (cm) | 156.44 (7.26) | 168.54 (7.07) | <0.001 |
| BMI (kg/m²) | 29.63 (6.23) | 31.40 (6.30) | 0.215 |
| Waist circumference (cm) | 93.65 (15.33) | 106.49 (14.19) | <0.001 |
| Hip circumference (cm) | 106.87 (12.09) | 104.76 (8.31) | 0.379 |
| Fat mass (kg) | 30.36 (12.42) | 31.88 (13.94) | 0.610 |
| Lean mass (kg) | 43.11 (9.83) | 49.02 (9.86) | 0.011 |
| Visceral adipose tissue (kg) | 1.60 (0.70) | 1.32 (0.89) | 0.136 |
| **Clinical parameters** | | | |
| Systolic blood pressure (mmHg) | 110.79 (24.15) | 129.08 (22.73) | 0.001 |
| Diastolic blood pressure (mmHg) | 70.02 (8.41) | 77.00 (9.08) | 0.001 |
| Obesity (BMI >= 30) (%) | 20 (46.5) | 17 (47.2) | 1.000 |

***Notes:*** *Values are presented as mean (standard deviation). Comparisons between groups were performed using the Wilcoxon rank-sum test, and proportions were compared using Chi-squared tests.*

## Mixed effects models

Mixed-effects model results are presented in Tables 2 and 3. In contrast, pointwise comparisons of leptin and adiponectin between meal types and obesity status at each postprandial time point are presented in S2–S5 Tables. In the linear mixed-effects models (Tables 2 and 3), fixed-effect coefficients represent the average change in adipokine concentration associated with each predictor, while accounting for repeated measurements within individuals through subject-level random intercepts.

The coefficient for time reflects the average rate of change in leptin (ng/mL) or adiponectin (μg/mL) concentration per minute during the postprandial period. Women and the high-fat meal were used as reference categories for sex and meal type, respectively. Random-effects estimates indicated between-subject variability, with adjusted intraclass correlation coefficients of 0.79 for leptin and 0.87 for adiponectin, supporting the use of subject-level random intercepts to account for repeated postprandial measurements. All fixed-effect estimates are presented with their corresponding 95% confidence intervals.

As summarized in Table 2, the leptin mixed-effects model included time, sex, BMI, meal type, and the interaction between BMI and meal composition as fixed effects, all of which were significant predictors of postprandial leptin concentrations. The negative time coefficient ($\beta = -0.004$) indicates a modest overall decline in leptin concentrations across the postprandial period. Men had lower leptin concentrations than women ($\beta = -26.05$), while higher BMI was associated with higher leptin concentrations ($\beta = 2.34$). In addition, the positive β coefficient for the high-carbohydrate meal ($\beta = 5.93$) indicates higher leptin concentrations compared with the high-fat meal. Still, the negative BMI x high-carbohydrate meal

**Table 2. Mixed-effects models evaluating postprandial leptin (n = 79).**

| Leptin ~ Time + Sex + BMI*Meal Type + (1|ID) | β | Standard Error | 95% CI | p-value |
|---|---|---|---|---|
| Time | −0.004 | 0.001 | −0.0078, −0.0005 | 0.03 |
| Men | −26.05 | 2.85 | −31.64, −2.05 | < 0.001 |
| BMI | 2.34 | 0.23 | 1.90, 2.80 | < 0.001 |
| Meal High-carbohydrate | 5.93 | 2.52 | 0.99, 10.9 | 0.02 |
| BMI: Meal High-carbohydrate | −0.23 | 0.008 | −7.03, −0.39, | 0.005 |
| Marginal R² | 62% | | | |
| Conditional R² | 92% | | | |

*For Meal Type, the "High in Fats" is the reference level. For Sex, "Female" is the reference level. Time was modelled as a continuous variable (minutes). β coefficients represent change per minute.*

**Table 3. Mixed-effects models evaluating postprandial adiponectin (n = 79).**

| Adiponectin ~ Time + Sex + BMI*Meal Type + (1|ID) | β | Standard Error | 95% CI | p-value |
|---|---|---|---|---|
| Time | −0.001 | 0.001 | −0.003, −0.001 | < 0.001 |
| Men | −3.3 | 0.8 | −5.02, −1.52 | < 0.001 |
| BMI | −2.32 | 0.72 | −0.37, −0.09 | 0.001 |
| Meal High-carbohydrate | −5.0 | 20.8 | −1.65, 0-64 | 0.38 |
| BMI: Meal High-carbohydrate | −0.18 | 0.01 | −0.02, 0.05 | 0.32 |
| Marginal R² | 24% | | | |
| Conditional R² | 90% | | | |

*For Meal Type, the "High in Fats" is the reference level. For Sex, "Female" is the reference level. Time was modelled as a continuous variable (minutes). β coefficients represent change per minute.*

interaction (β = −0.23) suggests that the BMI-related increases in leptin were weaker after the high-carbohydrate meal in comparison with the high-fat meal, consistent with a differential postprandial leptin response by meal composition. The model showed a marginal $R^2$ of 62% and a conditional $R^2$ of 92%, indicating that the fixed effects explained a substantial proportion of the variance in postprandial leptin concentrations, with additional variability accounted for by between-subject differences captured by the random intercepts.

As shown in Table 3, in the adiponectin mixed-effects model, sex and BMI were significant predictors of postprandial adiponectin concentrations. The negative time coefficient (β = −0.001) indicates a modest overall decline in adiponectin across the postprandial period. Men showed significantly lower adiponectin concentrations than women (β = −3.3). Additionally, higher BMI was independently associated with lower adiponectin, with a decrease of 2.32 µg/mL of adiponectin per 1 unit increase in BMI. In contrast, neither meal composition nor the BMI x meal interaction was significantly associated with adiponectin trajectories, indicating that postprandial adiponectin responses were not modified by meal composition. The model showed a marginal $R^2$ of 24% and a conditional $R^2$ of 90%, indicating that fixed effects explained a modest proportion of the variability in adiponectin levels, with most of the variance explained by subject differences captured by the random intercepts.

## Postprandial leptin and adiponectin responses

Figs 1, S1 and S2 illustrate the postprandial leptin and adiponectin trajectories, according to meal composition, sex, and BMI.

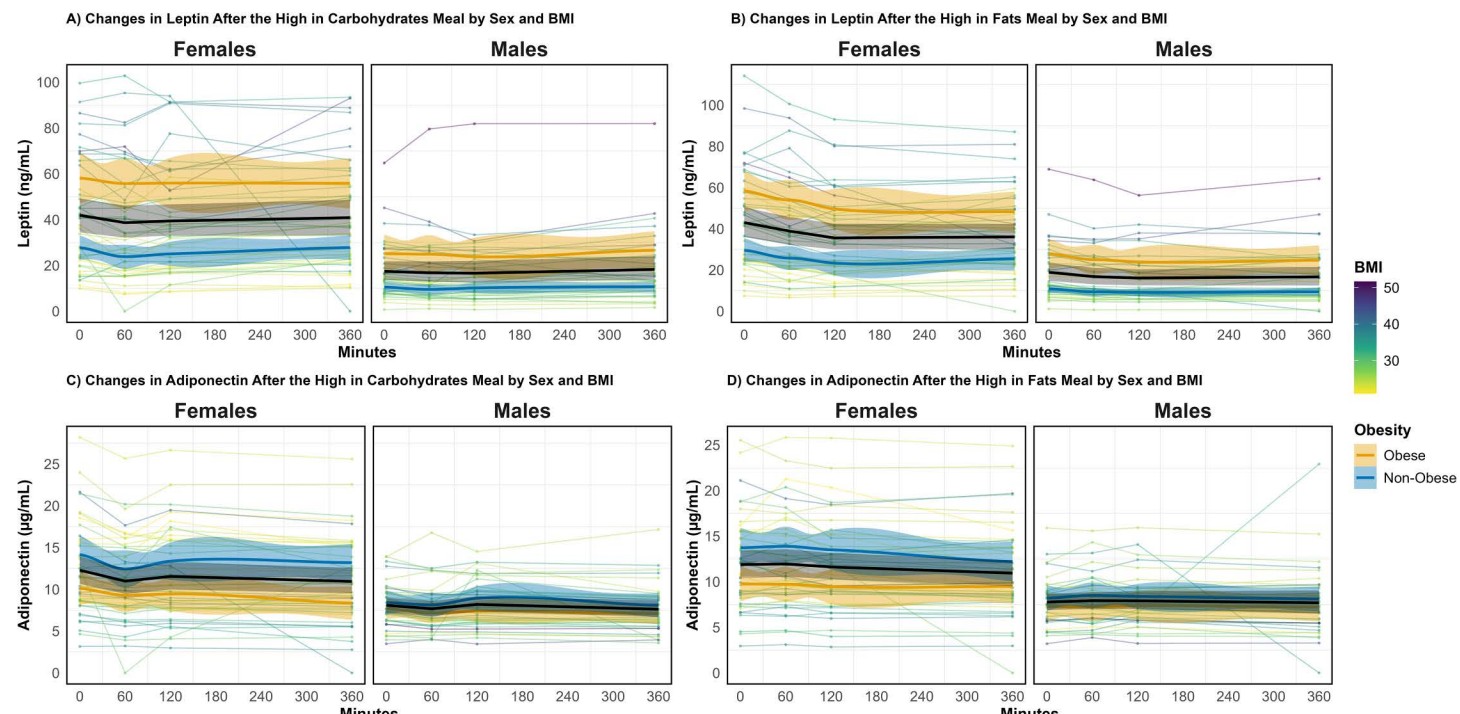

**Fig 1. Postprandial leptin and adiponectin trajectories by sex and BMI status. (A)** Leptin response after the high-carbohydrate meal. **(B)** Leptin response after the high-fat meal. **(C)** Adiponectin response after the high-carbohydrate meal. **(D)** Adiponectin response after the high-fat meal. In each panel, females are shown in the left sub-panel and males in the right sub-panel. Thick lines are LOESS smooths for non-obese (blue) and obese (orange) participants with 95% confidence bands; the black line is the overall smooth. Thin lines show individual trajectories, colored by BMI.

For leptin, after the high-carbohydrate meal (Figs 1A and S1), concentrations showed modest postprandial fluctuations, especially among women, while remaining higher in participants with obesity throughout the postprandial period. In contrast, the high-fat meal (Figs 1B and S1), leptin concentrations declined over time, with a more pronounced reduction among participants with obesity, particularly women. This contrast between panels A and B reflects the significant BMI x meal interaction in the mixed-effects model (Table 2). This finding indicates that the positive association between BMI and leptin concentration was attenuated after the high-fat meal.

For adiponectin (Table 3), sex and BMI were the main determinants of postprandial concentrations, with lower levels observed in men and in individuals with higher BMI. In contrast to leptin, neither meal type nor the interaction with BMI was significantly associated with adiponectin trajectories. Consistent with the model findings, Figs 1C, 1D and S2 illustrate relatively stable adiponectin concentrations throughout the postprandial period following both the high-carbohydrate and high-fat meals. Across both meal conditions, adiponectin levels remained consistently lower among those participants with obesity. These findings suggest that, while leptin dynamics are more sensitive to meal composition and to BMI status, adiponectin responses are primarily shaped by BMI and sex, with minimal influence of meal type.

When comparing meal types, no significant differences in leptin concentrations were observed at any postprandial time point between the high-carbohydrate and high-fat meals (S2 Table and S1 Fig). In contrast, when stratified by obesity status, participants with obesity consistently exhibited higher leptin concentrations than those without obesity across all time points, irrespective of meal type (S3 Table and S1 Fig).

Similarly, adiponectin concentrations did not differ significantly between the high-carbohydrate and high-fat meals at any postprandial time point (S4 Table and S2 Fig). When stratified by obesity status, participants without obesity had significantly higher adiponectin concentrations than those with obesity at most time points for both meal types (S5 Table). Differences were more consistent after the high-carbohydrate meal, being significant at all four time points. In contrast, after the high-fat meal, the difference was not significant at 360 min (p = 0.081) (S5 Table).

Together, these results indicate that while meal composition alone did not significantly influence postprandial leptin or adiponectin concentrations, obesity status and sex were key determinants of circulating levels. Leptin showed greater sensitivity to obesity with postprandial trajectories that differed by meal composition, as reflected by a significant BMI x meal interaction. In particular, the BMI-related increase in leptin was attenuated following the high-fat meal.

## Discussion

This study evaluated the postprandial responses of adiponectin and leptin over six hours after ingestion of standardized meals in adults with and without obesity. No significant differences between the high-carbohydrate and high-fat meals were observed at individual time points (0, 120, and 360 min). However, mixed- effects models demonstrated that sex, BMI, and meal composition were significant predictors of adipokine trajectories (Tables 2 and 3).

In the fasting state, our findings are consistent with previous reports showing higher concentrations of adiponectin and leptin in women compared with men [11]. Biological factors, including greater subcutaneous adipose tissue,– the primary source of leptin, and the influence of estrogens on adipokine regulation, may explain this pattern [12]. Women in our study also presented a more favorable lipid profile, with lower triglycerides and higher levels of HDL-C. In contrast, men showed higher glucose levels, blood pressure, and waist circumference, reflecting a more central fat distribution. These observations are consistent with prior evidence highlighting the role of fat distribution and hormonal milieu in shaping adipokine secretion and cardiometabolic risk [13,14].

We further observed that obesity was consistently associated with higher leptin and lower adiponectin concentrations across all postprandial time points, reinforcing their value as biomarkers of adiposity and metabolic dysfunction [5]. Although pointwise comparisons did not reveal significant differences between the high-carbohydrate and high-fat meals, trajectory analysis with mixed models indicated that meal composition shaped the overall leptin response. It is important to distinguish between these two levels of inference: the mixed-effects models capture overall trajectories across all time

points simultaneously, whereas pointwise Wilcoxon comparisons assess differences at each time point in isolation. The absence of significant pairwise differences at discrete time points does not contradict the model-based findings; rather, it reflects the statistical power of trajectory modeling to detect gradual, sustained effects that may not reach significance at any single time point. Specifically, after the high-fat meal, leptin concentrations showed a gradual decline over the six hours. In contrast, adiponectin concentrations exhibited only a modest, non-significant time-trend, consistent with the absence of a significant meal or BMI×meal effect in the adiponectin mixed-effects model. For each one-unit increase in BMI, the postprandial response was attenuated by 0.23 ng/mL following the high-fat meal compared with the high-carbohydrate meal, suggesting a blunted leptin response to dietary fat with increasing adiposity. In contrast, the absence of a significant BMI x meal interaction for adiponectin indicates that the relationship between BMI and adiponectin concentrations was not modified by meal composition.

This pattern suggests that while adipokines primarily reflect chronic adiposity, leptin, but not adiponectin, exhibits short-term, meal-dependent modulation, particularly dietary fat, in the postprandial period. While elucidating the mechanistic insights behind the observed changes in adiponectin and leptin falls beyond the scope of our paper and reach of our methods, we think it is reasonable to say that these results provide nuance to the common perception of leptin and adiponectin as "steady state" signals. The well-documented interplay between these two hormones and other "acute-phase" metabolic players, such as CCK, GLP-1, and insulin, within different physiological contexts, such as excessive adiposity and or inflammation [15] makes it reasonable to expect differential effects with differential health implications [16,17].

Other studies have generally reported minimal postprandial variation in leptin, supporting the hypothesis that leptin primarily reflects overall adiposity rather than acting as a rapidly responsive hormone to meal ingestion [18]. However, our observation of an acute decline after the high-fat meal challenges the classical view of leptin as an exclusively long-term signal of energy stores. It suggests that, under specific dietary conditions, leptin trajectories can be modified within a few hours.

Our findings are partly consistent with previous postprandial studies. Larsen et al.[19] reported a significant postprandial decline in leptin after a high-fat meal in normal-weight individuals, whereas this response was attenuated in participants with obesity, suggesting impaired leptin regulation in the obese state. Using longitudinal mixed-effects models, our results extend these observations by showing that meal composition modifies the association between BMI and leptin trajectories. Notably, the positive association between BMI and leptin concentrations was attenuated after the high-fat meal, indicating a greater postprandial decline in leptin with increasing adiposity. Together, these findings suggest that a diet high in fats may exacerbate alterations in leptin dynamics among individuals with obesity, even in the absence of significant pointwise differences.

The descriptive trend toward lower adiponectin concentrations after the high-fat meal (which did not reach statistical significance in the mixed-effects model), is consistent with previous reports of transient reductions following high-fat meals and concomitant postprandial lipemia [20]. However, in our study, meal composition did not emerge as a significant predictor in the mixed-effects model, and pointwise comparisons likewise showed no significant differences between meal types, indicating that any meal-related change in adiponectin was small in magnitude. Instead, postprandial adiponectin concentrations were primarily determined by sex and BMI, remaining consistently lower among participants with obesity across the postprandial period.

In this context, English et al.[21] evaluated the acute postprandial adiponectin response following a mixed meal in lean and obese individuals. Similarly, they reported significantly lower fasting adiponectin concentrations in participants with obesity, which were inversely associated with BMI and insulin resistance. However, in contrast to our findings, they observed a marked and transient postprandial increase in adiponectin among obese participants, peaking at approximately 60 minutes and remaining elevated up to 180 minutes, while no significant postprandial changes were observed in lean individuals. These discrepancies may be explained by differences in meal composition, study design, and analytical approaches, especially the use of longitudinal mixed-effects models in our study.

Finally, another study suggests that dietary fat quality may influence adiponectin regulation, with variable effects of different fatty acid types on adiponectin gene expression and circulating levels [22]. Nevertheless, within the context of our standardized acute challenges, adiponectin trajectories were not significantly modified by meal or by the BMI x meal interaction. Supporting BMI and sex as the main determinants of postprandial adiponectin in the present study.

In this study, linear mixed-effects models were the primary analytical approach to evaluate postprandial adipokine responses, as they allow the simultaneous modeling of longitudinal trajectories across all time points while accounting for within-subject correlation due to repeated measurements over time. While Wilcoxon rank-sum tests were used as complementary, exploratory analyses to assess between-group differences at individual postprandial time points, the absence of statistically significant differences at specific time points does not contradict the mixed-effects model findings, which reflect average postprandial trajectories and interactions over the entire observation period. The explanatory power of the mixed-effects models, as reflected by the marginal and conditional $R^2$ values, provides additional insight into the determinants of postprandial adipokine responses. For leptin, the high marginal $R^2$ indicates that fixed effects (including sex, BMI, and meal composition) explain a substantial proportion of the observed variability, and the strong influence of adiposity and dietary context on leptin postprandial dynamics.

In contrast, adiponectin models had lower marginal $R^2$ values, indicating that fixed predictors accounted for a more modest proportion of total variance. The marked difference between marginal and conditional $R^2$ suggests that the adiponectin variability is attributable to individual-level factors not fully captured by sex, BMI, or meal composition. In other words, while measurable phenotypic and dietary factors more strongly drive leptin responses, adiponectin levels appear to be more tightly regulated and less sensitive to acute nutritional modulation. Key strengths of our study include its cross-sectional design with two standardized meals, the assessment of repeated time points over six hours, and the use of mixed-effects models, which captured dynamic responses and accounted for a substantial proportion of inter-individual variability. However, limitations should be acknowledged. First, participants previously diagnosed with diabetes were excluded as an a priori design decision in order to characterize postprandial leptin and adiponectin responses in adults with chronic metabolic disease, beyond overweight and obesity. While this approach reduced metabolic heterogeneity and potential confounding related to glucose-lowering treatments, it may limit the generalizability of the findings of the study and introduce collider bias, particularly by the high incidence of coexistence of obesity and diabetes in the Mexican population. Second, these acute responses may not be directly generalizable to habitual dietary patterns, and the absence of additional biomarkers, particularly inflammatory markers, limited mechanistic insights. In addition, postprandial measurements of other key hormones, such as insulin and ghrelin or glucose, were not available. Finally, other lifestyle factors known to influence adipokine concentrations,– including physical activity, sleep, and stress,– were not assessed and may have contributed to residual variability.

Our results underscore the importance of postprandial dynamics beyond isolated fasting measurements. The progressive reductions in leptin, particularly among individuals with obesity, could potentially influence satiety-related pathways, although this hypothesis requires confirmation in studies directly assessing appetite and energy intake. These results suggest that dynamic postprandial phenotypes could be valuable markers for further characterizing patients with metabolic dysfunction. Considering meal composition and individual characteristics such as sex and adiposity may improve our understanding of adipokine regulation and its role in energy balance. Future studies incorporating longer-term dietary interventions and other biomarkers, including inflammatory and satiety-related pathways, postprandial glucose-insulin dynamics, and intestinal hormones, are needed to clarify the clinical implications of these hormonal dynamics.

## Conclusions

We demonstrated that although adiponectin and leptin concentrations showed minimal absolute changes after food intake, their postprandial trajectories were significantly influenced by individual characteristics and meal composition. These results highlight the relevance of dynamic postprandial phenotypes as early markers of metabolic dysfunction.

## Supporting information

**S1 Table. Standardized meals composition.**
(DOCX)

**S2 Table. Postprandial leptin concentrations by meal type.**
(DOCX)

**S3 Table. Postprandial leptin concentrations according to obesity status.**
(DOCX)

**S4 Table. Postprandial adiponectin concentrations by meal type.**
(DOCX)

**S5 Table. Postprandial adiponectin by obesity status and meal type.**
(DOCX)

**S1 Fig. Postprandial leptin distribution by time, sex, and obesity status.**
(PNG)

**S2 Fig. Postprandial adiponectin distribution by time, sex, and obesity status.**
(PNG)

## Author contributions

**Conceptualization:** Fabiola Del Razo-Olvera, Carlos Aguilar-Salinas, Adrian Soto-Mota.

**Data curation:** Fabiola Del Razo-Olvera, Ana Teresa Arias-Marroquín, Mayra Martínez-Fajardo, Wendolyn Elideth Davila-Olmedo, Ivette Cruz Bautista, Donaji Verónica Gomez-Velasco, Gabriela Alejandra Galan-Ramirez, Liliana Juárez-Martínez, Luz Elizabeth Guillén-Pineda, Rodrigo Otero-Otero, Daniel Elias-Lopez, Alexandro José Martagón-Rosado, Adrian Soto-Mota.

**Formal analysis:** Fabiola Del Razo-Olvera, Ana Teresa Arias-Marroquín, Adrian Soto-Mota.

**Funding acquisition:** Fabiola Del Razo-Olvera, Carlos Aguilar-Salinas.

**Investigation:** Fabiola Del Razo-Olvera, Ana Teresa Arias-Marroquín, Mayra Martínez-Fajardo, Wendolyn Elideth Davila-Olmedo, Ivette Cruz Bautista, Donaji Verónica Gomez-Velasco, Gabriela Alejandra Galan-Ramirez, Liliana Juárez-Martínez, Luz Elizabeth Guillén-Pineda, Angelina Lopez Estrada, Rodrigo Otero-Otero, Daniel Elias-Lopez, Alexandro José Martagón-Rosado, Adrian Soto-Mota.

**Methodology:** Fabiola Del Razo-Olvera, Wendolyn Elideth Davila-Olmedo, Daniel Elias-Lopez, Adrian Soto-Mota.

**Project administration:** Fabiola Del Razo-Olvera, Ana Teresa Arias-Marroquín, Wendolyn Elideth Davila-Olmedo, Ivette Cruz Bautista, Donaji Verónica Gomez-Velasco, Gabriela Alejandra Galan-Ramirez, Alexandro José Martagón-Rosado.

**Resources:** Fabiola Del Razo-Olvera.

**Software:** Adrian Soto-Mota.

**Supervision:** Fabiola Del Razo-Olvera, Carlos Aguilar-Salinas, Adrian Soto-Mota.

**Validation:** Adrian Soto-Mota.

**Visualization:** Ana Teresa Arias-Marroquín, Adrian Soto-Mota.

**Writing – original draft:** Fabiola Del Razo-Olvera, Ana Teresa Arias-Marroquín, Adrian Soto-Mota.

**Writing – review & editing:** Fabiola Del Razo-Olvera, Ana Teresa Arias-Marroquín, Mayra Martínez-Fajardo, Wendolyn Elideth Davila-Olmedo, Ivette Cruz Bautista, Donaji Verónica Gomez-Velasco, Gabriela Alejandra Galan-Ramirez, Liliana Juárez-Martínez, Luz Elizabeth Guillén-Pineda, Angelina Lopez Estrada, Rodrigo Otero-Otero, Daniel Elias-Lopez, Alexandro José Martagón-Rosado, Carlos Aguilar-Salinas, Adrian Soto-Mota.

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
