## [Decision Letter · Decision Letter 0]

18 Dec 2025

PONE-D-25-57507Postprandial Response of Leptin and Adiponectin to Standardized High-Carbohydrate and High-Fat Meals in Adults.PLOS One

Dear Dr. Soto-Mota, Thank you for submitting your manuscript to PLOS ONE. Four external referees have evaluated it, and the consensus is that it is of interest to the journal; however, there are several issues that merit a major revision. Additionally, I have comments the authors should consider to increase the strength of their conclusions. Please submit your revised manuscript by Feb 01 2026 11:59PM. If you will need more time than this to complete your revisions, please reply to this message or contact the journal office at plosone@plos.org. Please include the following items when submitting your revised manuscript:

We look forward to receiving your revised manuscript.

Kind regards,

Neftali Eduardo Antonio-Villa, MD PhD

Academic Editor

PLOS One

Journal Requirements:

3. Please note that your Data Availability Statement is currently missing the DOI/accession number of each dataset OR a direct link to access each database. If your manuscript is accepted for publication, you will be asked to provide these details on a very short timeline. We therefore suggest that you provide this information now, though we will not hold up the peer review process if you are unable.

Additional Editor Comments:

Therefore, we invite you to submit a revised version of the manuscript that addresses the points raised during the review process.

Please include a STROBE checklist that best adheres to the study design the authors used.The phrase “The institutional ethics committee approved the study.” is redundant, considering that the authors previously stated: “The institutional Ethics in Human Research Committee assessed and approved the protocol (reference number 3722).”A potential limitation of the study would be the exclusion of individuals living with diabetes, as this population also tends to have differential secretion of leptin and adiponectin, particularly in the elderly population (PMID: 33411633). This could prompt collider bias given the combined prevalence of diabetes and obesity in the Mexican population.Leptin seems to have a skewed distribution (given the high SD presented in Table 1). In this regard, did the residuals of the mixed-effects linear regression model fulfill a normal distribution?Consider including the 95% confidence intervals of the coefficients reported in Table 2 and Table 3. These could be obtained using the Wald approximation or a semiparametric bootstrap. The **lme4** R package further expands these approximations: https://cran.r-project.org/web/packages/lme4/vignettes/lmer.pdfAdditionally, please specify the fixed effects (time, sex, BMI, meal type) and random effects (individual) used in the models.

Minor comments

In Table 2, please consider changing the labels to “Men” instead of only “Sex” and “Low-carbohydrate meal” instead of “Meal type.”In figures, please be consistent with the labels “Men” and “Women.”Table 4 was not appended in the document.I believe there is a typo in lines 200 and 240.

Reviewer's Responses to Questions

**Comments to the Author**

1. Is the manuscript technically sound, and do the data support the conclusions?

Reviewer #1: Partly

Reviewer #2: Yes

Reviewer #3: Yes

Reviewer #4: Yes

2. Has the statistical analysis been performed appropriately and rigorously? 

Reviewer #1: Yes

Reviewer #2: Yes

Reviewer #3: Yes

Reviewer #4: Yes

3. Have the authors made all data underlying the findings in their manuscript fully available?

Reviewer #1: No

Reviewer #2: Yes

Reviewer #3: Yes

Reviewer #4: Yes

4. Is the manuscript presented in an intelligible fashion and written in standard English?

Reviewer #1: Yes

Reviewer #2: Yes

Reviewer #3: Yes

Reviewer #4: Yes

5. Review Comments to the Author

Reviewer #1: Manuscript Title: Postprandial Response of Leptin and Adiponectin to Standardized High-Carbohydrate and High-Fat Meals in Adults.

Manuscript ID: PONE-D-25-57507

Overall Recommendation: Major Revision

Overall Evaluation:

This study employed a crossover design to investigate the dynamic changes of leptin and adiponectin following standardized high-carbohydrate and high-fat meals, considering the influence of obesity status and sex. The research enrolled 79 adults (including obese and non-obese individuals) who underwent two standardized mixed-meal tolerance tests. Through mixed-effects model analysis, the study found that sex, BMI, and meal composition significantly affected the postprandial trajectories of leptin and adiponectin: both adipokines gradually decreased after a high-fat meal, while their concentrations remained relatively stable after a high-carbohydrate meal.

The topic of this study is relevant, addresses a knowledge gap regarding postprandial adipokine dynamics, and has potential implications for metabolic health. The use of mixed-effects models to capture dynamic responses is a major strength. All main tables, supplementary tables, and Figure 1 are now provided and accessible, greatly facilitating the review process. Figure 1 clearly illustrates the leptin and adiponectin dynamics in different sex and BMI groups after different meals, providing intuitive support for understanding the research results.

However, despite the completeness of all materials, the manuscript still presents several critical issues that must be addressed to meet PLOS One's publication standards, particularly concerning the rigor of methodological descriptions, the integration and interpretation of different statistical analysis results, and the depth of the discussion.

Major Comments:

Data Presentation and Interpretation Issues:

Confusing Citations of Main and Supplementary Tables: The text repeatedly misquotes Main Tables 2 and 4 (which are the primary tables presenting mixed-effects model results) when discussing pointwise comparisons of postprandial leptin and adiponectin. The detailed data for these pointwise comparisons should be presented in Supplementary Tables 2, 3, 4, and 5. For example, lines 174-175: "When comparing meal types, no significant differences in leptin concentrations were observed at any postprandial time point between the high-carbohydrate and high-fat meals (Table 2)." should be corrected to cite Supplementary Table 2. Please carefully check and correct all table citations in the text to ensure the purpose and citation logic of main and supplementary tables are clear.

Detailed Explanation of Mixed-Effects Models (Tables 2 and 3): This remains a core issue, especially given that mixed models are a primary strength of this study. The meaning of their coefficients must be precisely elucidated in the main text:

β-value for "Sex": Please clarify whether it represents an average concentration difference (with females as reference) or a difference in time trends. Based on Table 1 and the negative β-value, this appears to represent an average level difference for males compared to females, which should be explicitly stated in the text.

β-value for "Meal Type" (Leptin: 5.93, Adiponectin: -5.0): Please clearly explain its meaning. Does it represent the average concentration difference for the high-carbohydrate meal relative to the high-fat meal (reference level), or a difference in their overall postprandial trajectories? Given that supplementary tables show no significant pointwise differences, but the mixed model indicates meal type as a significant predictor, this distinction is crucial and requires clear clarification.

Interpretation of "BMI : Meal Type" Interaction Term: For leptin, this interaction term is significant (β = -0.23, p = 0.005), which supports the conclusion that "obese participants exhibited a significantly greater postprandial decline after the high-fat meal." Please elaborate on the biological significance of this interaction effect in the results and discussion, specifically, how the difference in leptin response between the two meal types changes for every one-unit increase in BMI. For adiponectin, this interaction term is not significant, indicating no differential effect of BMI on different meal types, which should also be stated in the text.

Detailed Interpretation of Figure 1: Figure 1 clearly presents the results, but the description and interpretation of Figure 1 in the main text still need strengthening. Please elaborate on the key trends and differences presented in each panel of Figure 1 in the results section, and connect them with the findings from the mixed-effects models (especially interactions). For example, how can the trend that "obese participants exhibited a significantly greater postprandial decline after the high-fat meal" be observed from Figure 1?

Methodological Details:

Inconsistent Description of Study Design: Line 101 of the manuscript still describes it as "This cross-sectional study…", which contradicts the actual "crossover design with repeated measures." Please ensure the study design type is corrected to an accurate description and maintained consistently throughout the manuscript.

Washout Period and Meal Order: This information is still missing. A crossover study design requires an adequate washout period to prevent carryover effects between interventions. Please clearly state the interval between the two MMTTs (washout period) in the methods and justify its adequacy. Additionally, please state whether the order of meal administration was randomized and whether any sequence or period effects were assessed.

Detailed Information on Standardized Dinner: The methods section mentions that "all participants consumed a standardized dinner the evening before each test day." While Supplementary Table 1 provides details of the test meals, the composition of the standardized dinner (macronutrients, calories, specific foods) is equally important for controlling the metabolic state before testing. Please provide these details in the methods or supplementary materials.

Further Clarification of Statistical Analysis:

Clear Distinction Between Wilcoxon and Mixed Models: The manuscript needs to clarify the respective roles of the Wilcoxon pointwise comparisons (Supplementary Tables 2-5) and the linear mixed-effects models (Main Tables 2-3). Wilcoxon tests show differences at specific time points, while mixed models capture overall longitudinal trajectories and interactions. The discussion must clearly reconcile the results of these two analytical approaches, explaining why single time points might not show significant differences, while the overall trajectory is significantly influenced by predictors.

Random Slopes for Time: The current model only includes "random intercepts for participants." Please consider whether "random slopes for time" should be included to account for individual differences in postprandial trends. If not included, please provide a clear justification.

Utilization of R² Values: The marginal R² and conditional R² values of the model have been reported, indicating good explanatory power. Please utilize these R² values more deeply in the discussion to elaborate on the extent to which the selected predictors (sex, BMI, meal composition) explain the variance in leptin and adiponectin levels.

Elaboration of Results and Discussion:

Integration and Logical Coherence of Core Findings: There is a contradiction between the conclusions in the abstract and results section regarding "postprandial leptin and adiponectin gradually decreased after a high-fat meal and remained stable after a high-carbohydrate meal" and "meal composition is a significant predictor," versus the statement in the main text (lines 174-175) that "When comparing meal types, no significant differences were observed at any postprandial time point." The authors must clearly explain this apparent contradiction in the results and discussion. For instance, emphasize that the mixed model captures trends over time or overall effects, while pointwise comparisons may not show significant instantaneous differences due to smaller effect sizes or insufficient statistical power.

Deepening Mechanistic Insights and Expanding References: The discussion section lacks in-depth exploration of potential mechanisms for the observed phenomena (especially the decrease in leptin and adiponectin after a high-fat meal). Please expand this section, incorporating existing literature to discuss possible physiological mechanisms, such as the effects of different macronutrients on insulin, glucose, gut hormones (e.g., CCK, GLP-1), as well as fatty acid types, inflammatory responses, etc. The current number of 13 references is relatively small; it is recommended to supplement with more comprehensive and relevant literature to support the depth of the background and discussion.

Clinical Significance and "Leptin Resistance": The discussion states that "a reduction in leptin, particularly in obese individuals, may impair satiety signals and contribute to excessive energy intake." Please further link this finding to the concept of "leptin resistance," explaining how these postprandial dynamics offer new perspectives for understanding energy imbalance in obesity.

Language and Formatting:

Language Correction: In the abstract, "Contrastingly, stable and did not change significantly after the high-carbohydrate meal" is still missing a subject (e.g., "their concentrations remained stable"). Please carefully check the entire manuscript to ensure fluent language, accurate expression, and correct all grammatical and spelling errors (e.g., "produces" in line 342 "All plots were produces using ggplot2." should be "produced").

Acronym Handling: Ensure all acronyms are defined upon their first appearance. "LCHF" in the list of acronyms is not directly relevant to the meal descriptions in this study; if not further discussed in the main text, it is suggested to remove it.

Table Units and Legends: Please carefully check the units for adiponectin in Supplementary Tables 4 and 5, currently displayed as "(mean ± SD, ng/mL)," which is likely a typo and should be µg/mL. Ensure all tables and figure legends are clear, self-consistent, and contain complete information.

Minor Comments:

Ethics Approval and Data Collection Dates: Lines 106-107 "Recruitment started on July 1st, 2022 and finished on July 31st, 2025" suggest the study is still ongoing. Please confirm whether data collection was completed at the time of manuscript submission. If so, please update the end date to reflect its completed status.

Reference Dates and Format: Dates such as "cited 2025 Sep 21" or "cited 2025 May 15" in the references are still future dates and should be corrected to actual citation or retrieval dates. PLOS One encourages the use of DOIs or PMCID. Please check the reference format to ensure full adherence to PLOS One's guidelines.

Lean Mass Data in Table 1: Table 1 shows that average lean mass for females (49.02 ± 9.86 kg) is higher than for males (43.11 ± 9.83 kg, p=0.011), which contradicts general knowledge for the adult population. Please re-verify these data. If the data are accurate, please provide a reasonable explanation in the discussion (e.g., specific characteristics of the study population, age distribution, body composition measurement methods) or highlight it as a unique finding of this study.

Data and Code Availability: The manuscript submission system indicates data availability as "No - some restrictions will apply," with the explanation that data belong to a biobank and are only accessible to qualified researchers. This is an acceptable exception; please ensure this data availability statement is explicitly included in the methods section or supplementary information of the main manuscript, along with contact information or a process for obtaining the data.

Reporting Guidelines: It is recommended that the authors refer to relevant reporting guidelines, such as STROBE (for observational studies) or STROBE-nut (for nutritional studies), when preparing the revised manuscript to ensure comprehensive and transparent reporting of the research.

Conclusion:

Now, with the complete manuscript content (including Figure 1 and all tables), a comprehensive evaluation is possible. The topic explored in this study is interesting, and valuable findings have been obtained through the use of mixed-effects models. However, to meet PLOS One's rigorous publication standards, particularly regarding the precision of methodological descriptions, the clear integration and in-depth interpretation of different statistical analysis results, and the depth of discussion on potential mechanisms, comprehensive revisions are still required. Please address all the comments above upon resubmission. If these issues can be adequately resolved, I believe this manuscript will make a valuable contribution to the field.

Reviewer #2: This article offers a novel perspective. Despite the relatively small sample size, it has preliminarily uncovered the phenomenon that leptin and adiponectin levels tend to decrease following a high - fat meal. Nevertheless, the study did not conduct an in - depth exploration of the underlying mechanisms. For instance, it remains unclear whether changes in key metabolic signals, such as postprandial insulin, free fatty acids, inflammatory factors, or gastrointestinal hormones, are involved in regulating this response. It is recommended that in the discussion section, possible pathways should be further hypothesized in conjunction with existing literature. Specifically, does high - fat intake result in a reduction in circulating levels by inhibiting leptin secretion or accelerating its clearance? Does postprandial lipemia impact the stability or detection accuracy of adiponectin? Supplementing the relevant experiments to further elucidate the aforementioned potential mechanisms would substantially strengthen the integrity and persuasiveness of the study. These mechanistic considerations will substantially enhance the scientific rigor and theoretical value of the article.

Reviewer #3: Summary: Del Razo-Olvera et al. measured leptin and adiponectin levels following two meals, one high-fat and one high-carbohydrate. They report differences between individuals based on sex, BMI and meal composition.

Overall comments: The manuscript was well written and easy to follow. The introduction and discussion sections included pertinent information, and the study design was thorough. There were only a few comments to improve reader comprehension as outlined below:

1. It would be helpful for the authors to add bar graphs to demonstrate postprandial declines in both leptin and adiponectin between sexes and obese states to more clearly represent differences. These would be in relation to figure 1 and allow for more direct comparisons visually between the different groups.

2. Table 4 is referenced in line 180 but not included in the data.

Reviewer #4: This study by Del Razo-Olvera and co-workers addresses the postprandial responses of two of the most paradigmatic hormones produced by adipose tissue (adipokines), central in the control of metabolic homeostasis (adiponectin) and our relationship with food (leptin), driving together, when defective, the development of lifestyle diseases. The question that they arise is, then, very important, and the study they developed, thus, pretty pertinent. However, it is not true that there is a lack of information regarding the postprandial fluctuations of these two well-known and extensively studied adipokines, or that their concentrations are unaffected by food intake. There is room for improvement in this regard, and quite a lot of very compelling studies dealing with the topic being neglected. The authors need to state crystal clear what’s supposed to be new and what’s confirmatory in their study, and leverage their results with the data of others researchers digging in the topic. Below, there are some other suggestions that the authors of this study can, hopefully, find useful, in order to boost their research results:

Additional points that might improve after revision:

1) Some typos and mistakes have been identified along the manuscript. Please, be careful we word usage.

2) Some references are missing/not very well placed/explained along the text. The authors neglect some key literature on the topic. Please, revise.

3) The plots provided (Fig.1c-d) are okay, but they do not really show the changes the authors claim in the text, apparently provided in supplementary tables. To better contextualising the information provided in text and overall picture, please include any relevant data as main article figures.

4) Some additional markers following the lead of leptin and adiponectin in response to these two dietary inputs may help in boosting the relevance and clinical interest of this study. I am thinking about post-prandial measures of glucose, insulin and ghrelin during the time-course, see to what extent they level adiponectin and leptin fluctuations.

5) Some authors have related the adiponectin-leptin ratio to meta-inflammation (e.g., PMID: 30813240, PMID: 34333093). Do the current results show anything in that line of though?

6) “These results highlight the relevance of dynamic postprandial phenotypes as early markers of metabolic dysfunction.”: Not sure what that means.

6. PLOS authors have the option to publish the peer review history of their article (what does this mean?). If published, this will include your full peer review and any attached files.

Reviewer #1: No

Reviewer #2: No

Reviewer #3: No

Reviewer #4: **Yes:**Francisco J Ortega

---

## [Author Response · Author response to Decision Letter 1]

27 Jan 2026

RESPONSE TO REVIEWERS

Title: Postprandial Response of Leptin and Adiponectin to Standardized High Carbohydrate and High-Fat Meals in Adults.

Manuscript ID: PONE-D-25-57507

Dear Dr. Antonio-Villa,

We sincerely appreciate the opportunity to revise and improve our manuscript entitled “Postprandial Response of Leptin and Adiponectin to Standardized High Carbohydrate and High-Fat Meals in Adults,” and for the time and effort invested by you and the reviewers in evaluating our work.

We have carefully reviewed all your comments and those of the reviewers. Below, we provide a response to each comment. To facilitate their review, all editorial-team comments are displayed in red. Our manuscript has been revised accordingly, and all changes made in the enclosed manuscript are indicated in highlighted text in the revised version. To facilitate reviewing these changes, we added the corresponding page numbers in our responses, too.

We hope this current version of our manuscript now meets the standards to be eligible for publication in your journal. Nonetheless, we would be happy to provide any additional information if required. Thank you once again for the time, effort, and consideration granted to our submission.

Dr. Adrian Soto-Mota, MD, PhD, FACP.

Chief of the Metabolic Diseases Research Unit

National Institute of Medical Science and Nutrition Salvador Zubirán

adrian.sotom@incmnsz.mx,

phone: +5255-5487-0900, ext: 6319

Editorial Comment:

If applicable, we recommend that you deposit your laboratory protocols in protocols.io to enhance the reproducibility of your results.

Response: Thank you for this recommendation. At this stage, we did not deposit our laboratory protocols in protocols.io, as the procedures used in this study are fully described in the Methods section. Any additional information regarding our methods can be obtained by emailing the corresponding author. We will adopt this suggestion in future projects.

Journal Requirements:

Response: We confirm that the corrected manuscript has been thoroughly reviewed and revised to fully adhere to PLOS ONE’s style and formatting standards, including file naming conventions. The new version of our manuscript was formatted using the official PLOS ONE style templates for the main body, title page, authors, and affiliations.

2. We note that the grant information you provided in the ‘Funding Information’ and ‘Financial Disclosure’ sections does not match. When you resubmit, please ensure that you provide the correct grant numbers for the awards you received for your study in the ‘Funding Information’ section.

Response: Thank you for the comment. The Funding Information and Financial Disclosure sections have been reviewed and corrected to ensure consistency.

3. Please note that your Data Availability Statement is currently missing the DOI/accession number of each dataset OR a direct link to access each database. If your manuscript is accepted for publication, you will be asked to provide these details on a very short timeline. We therefore suggest that you provide this information now, though we will not hold up the peer review process if you are unable.

Response: Thank you for pointing this out. Since this study is part of the Mexican Biobank for Metabolic Diseases (BIOMEM), we are not allowed to make the datasets immediately available. The Data Availability Statement has been revised to reflect this.

Response: Captions for all Supporting Information files have now been corrected at the end of the manuscript, and all corresponding in-text citations have been revised and modified to ensure consistency with the Supporting Information files. Page 20.

Response: Thank you for this clarification. All publications suggested by the reviewers have been carefully evaluated and, since they are relevant indeed, added to the manuscript where appropriate, in accordance with the reviewers’ guidance. Page 21.

Additional Editor Comments:

Please include a STROBE checklist that best adheres to the study design the authors used.

Response: A revised version of the manuscript addressing all points raised during the review process has been submitted. In addition, an appropriate STROBE checklist for the study design has been included as Supporting Information.

1. The phrase “The institutional ethics committee approved the study.” is redundant, considering that the authors previously stated: “The institutional Ethics in Human Research Committee assessed and approved the protocol (reference number 3722).”

Response: We amended our text accordingly. Page 6.

2. A potential limitation of the study would be the exclusion of individuals living with diabetes, as this population also tends to have differential secretion of leptin and adiponectin, particularly in the elderly population (PMID: 33411633). This could prompt collider bias given the combined prevalence of diabetes and obesity in the Mexican population.

Response: Thank you for this insightful comment. The exclusion of subjects with previously diagnosed diabetes was a prior design decision to characterize postprandial leptin and adiponectin responses in adults without chronic metabolic disease beyond overweight or obesity. This approach allowed us to reduce metabolic heterogeneity and potential confounding related to glucose-lowering treatments and diabetes duration. Nonetheless, we agree that this exclusion criterion may limit the generalizability of our findings and could introduce collider bias, and we included these points in our limitations section. Page 19.

3. Leptin seems to have a skewed distribution (given the high SD presented in Table 1). In this regard, did the residuals of the mixed-effects linear regression model fulfill a normal distribution?

Response: Thanks for this important statistical point. Although leptin concentration showed a right-skewed distribution, model assumptions were evaluated based on the mixed-effects model residuals. Residual diagnostics, including histograms and Q-Q plots, indicated approximate normality, with only minor deviations at the extremes. Therefore, the use of mixed-effects linear regression models was considered appropriate.

4. Consider including the 95% confidence intervals of the coefficients reported in Table 2 and Table 3. These could be obtained using the Wald approximation or a semiparametric bootstrap. The lme4 R package further expands these approximations: https://cran.r-project.org/web/packages/lme4/vignettes/lmer.pdf -

Response:

Thank you for the suggestion. We have added 95% confidence intervals for all fixed-effect coefficients to Tables 2 and 3, using the Wald approximation implemented in the lme4 package. The tables had been updated accordingly.

5. Additionally, please specify the fixed effects (time, sex, BMI, meal type) and random effects (individual) used in the models.

Response: Thank you for the suggestion. We have now explicitly specified the fixed effects (time, sex, BMI, and meal type) and the random effect (individual participant), and we have included mixed-effects models in the Statistical Analysis section. Page 8.

Minor comments

In Table 2, please consider changing the labels to “Men” instead of only “Sex” and “Low-carbohydrate meal” instead of “Meal type.”

Response: Thank you. We have replaced “Sex” with “Men” and “Meal type” with “Low-carbohydrate meal” in tables 2 and 3.

In figures, please be consistent with the labels “Men” and “Women.”

Response: Thank you. All figures have been reviewed and updated with consistent labels “Men” and “Women.”

Table 4 was not appended in the document.

Response: Thank you. This was a numbering error, which has now been corrected, and all table numbering has been updated accordingly.

I believe there is a typo in lines 200 and 240.

Response: Thank you. Lines 200 and 240 were carefully reviewed, and the likely typographical issues were corrected in the revised manuscript.

Major Comments:

Data Presentation and Interpretation Issues:

Confusing Citations of Main and Supplementary Tables: The text repeatedly misquotes Main Tables 2 and 4 (which are the primary tables presenting mixed-effects model results) when discussing pointwise comparisons of postprandial leptin and adiponectin. The detailed data for these pointwise comparisons should be presented in Supplementary Tables 2, 3, 4, and 5. For example, lines 174-175: "When comparing meal types, no significant differences in leptin concentrations were observed at any postprandial time point between the high-carbohydrate and high-fat meals (Table 2)." should be corrected to cite Supplementary Table 2. Please carefully check and correct all table citations in the text to ensure the purpose and citation logic of main and supplementary tables are clear.

Response: Thank you for this observation. We have revised all table citations throughout the manuscript and corrected them.

Detailed Explanation of Mixed-Effects Models (Tables 2 and 3): This remains a core issue, especially given that mixed models are a primary strength of this study. The meaning of their coefficients must be precisely elucidated in the main text:

Response: We agree that a clear interpretation of mixed-effects model coefficients is essential. We have revised this section and explained the meaning of each β coefficient in Tables 2 and 3. We have carefully reviewed all results to ensure full consistency between the main text and the main and supplementary tables. In addition, errors related to the specification of the meal reference category in the mixed-effects models were identified and corrected in Tables 2 and 3. Pages 10-11.

β-value for "Sex": Please clarify whether it represents an average concentration difference (with females as reference) or a difference in time trends. Based on Table 1 and the negative β-value, this appears to represent an average level difference for males compared to females, which should be explicitly stated in the text.

Response: The β coefficient for sex represents an average difference in adipokine concentrations between men and women, with women specified as the reference category, and does not reflect a difference in time trends. The negative β value, therefore, indicates lower overall postprandial leptin and adiponectin concentrations in men compared with women, independent of time and meal type. We have revised the Results section accordingly.

β-value for "Meal Type" (Leptin: 5.93, Adiponectin: -5.0): Please clearly explain its meaning. Does it represent the average concentration difference for the high-carbohydrate meal relative to the high-fat meal (reference level), or a difference in their overall postprandial trajectories? Given that supplementary tables show no significant pointwise differences, but the mixed model indicates meal type as a significant predictor, this distinction is crucial and requires clear clarification.

Response: The β coefficient for meal type in the mixed-effects models represents an average difference in adipokine concentrations across the entire postprandial period, adjusted for time, sex, BMI, and their interaction, with the high-carbohydrate meal specified as the reference category.

For leptin, the β-value indicates that, on average, postprandial leptin concentrations were higher following the high-fat meal compared with the high-carbohydrate meal, independent of time trends. This coefficient does not reflect differences in time-specific trajectories. The absence of significant pointwise differences between meals in the supplementary tables reflects the use of nonparametric comparisons at individual time points, whereas the mixed-effects model estimates an overall adjusted mean difference across measurements.

Interpretation of "BMI : Meal Type" Interaction Term: For leptin, this interaction term is significant (β = -0.23, p = 0.005), which supports the conclusion that "obese participants exhibited a significantly greater postprandial decline after the high-fat meal." Please elaborate on the biological significance of this interaction effect in the results and discussion, specifically, how the difference in leptin response between the two meal types changes for every one-unit increase in BMI. For adiponectin, this interaction term is not significant, indicating no differential effect of BMI on different meal types, which should also be stated in the text.

Response: We have revised both the results and discussion sections to explicitly interpret the BMI x meal type interaction. For leptin, the significant negative interaction term (β = -0.23, p = 0.005) indicates that for each 1 unit increase in BMI, the difference in postprandial leptin concentrations between the high-fat and high-carbohydrate meals decreases by 0.23 ng/mL, reflecting a greater postprandial decline in leptin after the high-fat meal as adiposity increases. For adiponectin, we explained that the BMI x meal interaction was not significant, indicating no differential effect of BMI on adiponectin responses across meal types.

Detailed Interpretation of Figure 1: Figure 1 clearly presents the results, but the description and interpretation of Figure 1 in the main text still need strengthening. Please elaborate on the key trends and differences presented in each panel of Figure 1 in the results section and connect them with the findings from the mixed-effects models (especially interactions). For example, how can the trend that "obese participants exhibited a significantly greater postprandial decline after the high-fat meal" be observed from Figure 1?

Response: The results section was revised to explicitly describe the key trends observed in each panel and to connect them with the mixed-effects model results, particularly the significant interaction between BMI and type of meal for leptin. In addition, Figure 1 was carefully reviewed, and the panel order was slightly modified to improve clarity, with men consistently shown on the left and women on the right across all panels. Pages 12-13

Methodological Details:

Inconsistent Description of Study Design: Line 101 of the manuscript still describes it as "This cross-sectional study…", which contradicts the actual "crossover design with repeated measures." Please ensure the study design type is corrected to an accurate description and maintained consistently throughout the manuscript.

Response: Thank you. We have revised the manuscript to ensure the cross-sectional design is consistently and clearly described throughout the text.

Washout Period and Meal Order: This information is still missing. A crossover study design requires an adequate washout period to prevent carryover effects between interventions. Please clearly state the interval between the two MMTTs (washout period) in the methods and justify its adequacy. Additionally, please state whether the order of meal administration was randomized and whether any sequence or period effects were assessed.

Response: Thank you for this comment. Participants completed two test visits within the same week because both interventions were designed as acute postprandial challenges, and to minimize changes in habitual diet and lifestyle that could confound results. We included these details in the methods section. Page 7.

Detailed Information on Standardized Dinner: The methods section mentions that "all participants consumed a standardized dinner the evening before each test day." While Supplementary Table 1 provides details of the test meals, the composition of the standardized dinner (macronutrients, calories, specific foods) is equally important for controlling the metabolic state be

---

## [Decision Letter · Decision Letter 1]

24 Feb 2026

PONE-D-25-57507R1Postprandial Response of Leptin and Adiponectin to Standardized High-Carbohydrate and High-Fat Meals in Adults.PLOS One

Dear Dr. Soto-Mota,

Thank you for submitting your manuscript to PLOS ONE. After careful consideration and review of the reports from three external referees, we believe the manuscript has potential for publication; however, several issues must still be addressed before we can proceed. We therefore invite you to submit a revised version that responds to the reviewers’ comments and any associated editorial recommendations.

We look forward to receiving your revised manuscript.

Kind regards,

Neftali Eduardo Antonio-Villa, MD PhD

Academic Editor

PLOS One

Journal Requirements:

Additional Editor Comments:

I commend the authors for the extensive work undertaken to address the comments raised by all referees. Please also attend to the remaining points outlined in Reviewer 1’s final considerations.

Reviewers' comments:

Reviewer's Responses to Questions

**Comments to the Author**

1. If the authors have adequately addressed your comments raised in a previous round of review and you feel that this manuscript is now acceptable for publication, you may indicate that here to bypass the “Comments to the Author” section, enter your conflict of interest statement in the “Confidential to Editor” section, and submit your "Accept" recommendation.

Reviewer #1: All comments have been addressed

Reviewer #3: All comments have been addressed

Reviewer #4: (No Response)

2. Is the manuscript technically sound, and do the data support the conclusions?

Reviewer #1: Yes

Reviewer #3: Yes

Reviewer #4: Partly

3. Has the statistical analysis been performed appropriately and rigorously? 

Reviewer #1: Yes

Reviewer #3: Yes

Reviewer #4: Yes

4. Have the authors made all data underlying the findings in their manuscript fully available?

Reviewer #1: Yes

Reviewer #3: Yes

Reviewer #4: No

5. Is the manuscript presented in an intelligible fashion and written in standard English?

Reviewer #1: Yes

Reviewer #3: Yes

Reviewer #4: Yes

6. Review Comments to the Author

Reviewer #1: Manuscript ID: PONE-D-25-57507R1

Title: Postprandial Response of Leptin and Adiponectin to Standardized High-Carbohydrate and High-Fat Meals in Adults

General assessment

This manuscript investigates the postprandial dynamics of leptin and adiponectin following two standardized mixed meals with different macronutrient compositions in adults with and without obesity. The topic is clinically relevant and fits well within the scope of PLOS ONE, as it addresses an understudied aspect of adipokine physiology under controlled postprandial conditions.

Overall, the study is methodologically sound, the analyses are appropriate, and the conclusions are generally supported by the data. The revised manuscript is clearly written and substantially improved in clarity and structure compared to the original submission. Importantly, the authors do not overstate novelty and frame their conclusions conservatively, which is appropriate for this journal.

Based on PLOS ONE publication criteria, I believe the manuscript is suitable for publication pending minor to moderate revisions, primarily aimed at clarifying methodology, statistical interpretation, and data availability.

Major comments

1.Study design and crossover clarification

The manuscript states that participants completed two standardized mixed-meal tolerance tests with different macronutrient compositions. However, it is not fully clear whether the study followed a randomized crossover design or whether the order of meal administration was fixed.

Please clarify:

a Whether meal order was randomized or counterbalanced.

b The washout period between test days.

c:Whether potential order or carryover effects were assessed or controlled for.

Clarification is important because postprandial hormonal responses may be influenced by prior dietary exposure.

2.Interpretation of meal composition effects

While the mixed-effects models identify meal composition as a significant predictor of leptin trajectories, pointwise comparisons between meals at individual time points were largely non-significant.

The Discussion should more clearly distinguish between:

a:Statistical significance in trajectory modeling versus

b:Lack of differences at discrete postprandial time points.

At present, the manuscript occasionally implies stronger meal effects than are directly supported by the pairwise comparisons. A more nuanced interpretation would improve scientific rigor and transparency.

3.Adiponectin findings and model interpretation

For adiponectin, the mixed-effects model indicates that meal composition and BMI-by-meal interactions were not significant predictors, yet the Discussion suggests a modest decline after the high-fat meal.

Please clarify whether this decline reflects:

a :A statistically supported trajectory effect, or

b:A descriptive trend without formal statistical support.

If the latter, the language should be softened to avoid overinterpretation.

4.Data availability statement

The Data Availability section indicates that data are part of Mexico’s BIOMEM biobank and are available upon meeting criteria for access.

Please specify:

a:Whether aggregated or de-identified data underlying the figures and tables (e.g., summary statistics) can be made publicly available.

b:The process and contact information for requesting access.

This clarification is necessary to ensure compliance with PLOS ONE’s data availability requirements.

Minor comments

Statistical reporting

Please specify whether residual diagnostics were examined for the linear mixed-effects models.

Clarify whether time was modeled as a continuous or categorical variable.

Terminology

In several places, “stable” is used without an explicit statistical reference. Consider replacing with “did not change significantly” for precision.

Figures

Figures are clear and informative. However, the color gradient representing BMI may be difficult to interpret for readers with color vision deficiencies. Consider whether an alternative or simplified representation is feasible.

Typographical and stylistic issues

Minor grammatical inconsistencies remain (e.g., spacing, punctuation around units). A final careful proofreading is recommended.

Ethical considerations

The study includes appropriate ethical approval and informed consent statements. No ethical concerns were identified.

Reviewer #3: No additional comments. The authors have addressed all relevant comments from reviewers and the editor.

Reviewer #4: The manuscript has improved as much as it could without further experimentation. This reviewer has no further comments.

7. PLOS authors have the option to publish the peer review history of their article (what does this mean?). If published, this will include your full peer review and any attached files.

Reviewer #1: No

Reviewer #3: No

Reviewer #4: No

---

## [Author Response · Author response to Decision Letter 2]

18 Mar 2026

Dear Dr. Antonio-Villa

Thank you for the opportunity to revise our manuscript and for the reviewers' careful

evaluation. We appreciate the constructive feedback and are pleased that the reviewers

indicated that their previous comments have been adequately addressed.We carefully

reviewed the manuscript once more to ensure clarity and consistency throughout the text.

We provide a response to each comment below. All changes made in the manuscript are

indicated in highlighted text in the revised version, and the corresponding page numbers are

provided in our responses.

We hope that the revised manuscript is now suitable for publication in PLOS ONE. Please let

us know if any additional information is required.

Thank you again for your consideration.

Dr. Soto-Mota

Major comments

1. Study design and crossover clarification

The manuscript states that participants completed two standardized mixed-meal tolerance

tests with different macronutrient compositions. However, it is not fully clear whether the

study followed a randomized crossover design or whether the order of meal administration

was fixed.

Please clarify:

a Whether meal order was randomized or counterbalanced.

Response: We thank the reviewer for this comment. All participants completed two

standardized mixed-meal tolerance tests with different macronutrient compositions (highcarbohydrate and high-fat). The order of meal administration was randomly assigned for each

participant to reduce potential order effects. This information has now been clarified in the

Methods section. Page 7

b The washout period between test days.

Response: Participants completed the two mixed-meal tolerance tests on separate visits

within the same week, with an interval of 1 to 7 days between visits. Since the outcomes

assessed were acute postprandial responses, a formal washout period was not implemented.

Page 7

c: Whether potential order or carryover effects were assessed or controlled for.

Clarification is important because postprandial hormonal responses may be influenced by

prior dietary exposure.

Response: Because of the short-term nature of the outcomes, the risk of carryover effects

from the previous meal challenge was considered minimal. Page 7

2. Interpretation of meal composition effects

While the mixed-effects models identify meal composition as a significant predictor of leptin

trajectories, pointwise comparisons between meals at individual time points were largely

non-significant.

The Discussion should more clearly distinguish between:

a:Statistical significance in trajectory modeling versus

b:Lack of differences at discrete postprandial time points.

At present, the manuscript occasionally implies stronger meal effects than are directly

supported by the pairwise comparisons. A more nuanced interpretation would improve

scientific rigor and transparency.

Response: The mixed-effects models evaluate the overall postprandial trajectory across all

time points simultaneously, offering greater statistical power to detect gradual, sustained

effects. In contrast, the pointwise Wilcoxon comparisons test for differences at each

individual time point in isolation, without borrowing information across the entire time

series. The lack of significant pairwise differences at discrete time points therefore does not

contradict the model-based results; it reflects the inherently lower power of single-timepoint

comparisons to detect gradual trends. We have revised the Discussion to make this distinction

explicit and to ensure that language throughout is consistent with the level of evidence

provided by each analytical approach. Specifically, we have softened statements that implied

stronger meal effects than were directly supported by the pairwise comparisons.

Change made: Added a paragraph in the Discussion: "It is important to distinguish between

these two levels of inference: the mixed-effects models capture overall trajectories across all

time points simultaneously, whereas pointwise Wilcoxon comparisons assess differences at

each individual time point in isolation. The absence of significant pairwise differences at

discrete time points does not contradict the model-based findings; rather, it reflects the greater

statistical power of trajectory modeling to detect gradual, sustained effects that may not reach

significance at any single time point." Additionally, the phrase 'shaped the overall hormonal

response' was revised to 'shaped the overall leptin response' to avoid implying a global effect

on adiponectin that was not observed. Page 15.

3. Adiponectin findings and model interpretation

For adiponectin, the mixed-effects model indicates that meal composition and BMI-by-meal

interactions were not significant predictors, yet the Discussion suggests a modest decline

after the high-fat meal.

Please clarify whether this decline reflects:

a :A statistically supported trajectory effect, or

b:A descriptive trend without formal statistical support.

If the latter, the language should be softened to avoid overinterpretation.

Response: Thank you, the comment is correct. The mixed-effects model did not identify meal

composition or the BMI × meal interaction as significant predictors of adiponectin

trajectories. The observed decline after the high-fat meal is a descriptive trend visible in the

figures and consistent with prior literature, but it is not formally supported by the statistical

model. We have revised the relevant sentences in the Discussion to make this explicit, with

phrasing that clearly frames it as a descriptive, non-significant observation.

Change made: The phrase "The modest decline in adiponectin observed after the high-fat

meal" was revised to "The descriptive trend toward lower adiponectin concentrations after

the high-fat meal (which did not reach statistical significance in the mixed-effects model)."

Similarly, "adiponectin concentrations exhibited only modest time-related changes,

consistent with the mixed-model results" was revised to "a modest, non-significant timerelated trend, consistent with the absence of a significant meal or BMI×meal effect in the

adiponectin mixed-effects model." Page 15.

4. Data availability statement

The Data Availability section indicates that data are part of Mexico’s BIOMEM biobank and

are available upon meeting criteria for access.

Please specify:

a:Whether aggregated or de-identified data underlying the figures and tables (e.g., summary

statistics) can be made publicly available.

b:The process and contact information for requesting access.

This clarification is necessary to ensure compliance with PLOS ONE’s data availability

requirements.

Response:

We appreciate the reviewer attention to the PLOS Data Availability Policy. To clarify data

access procedures, we have revised the Data Availability Statement to specify that the dataset

belongs to Mexico's Biobank for Medical Diseases (BIOMEM) and that access is subject to

institutional ethical and legal restrictions regarding participant privacy. For this reason,

neither individual-level nor aggregated datasets can be publicly available. However, deidentified data supporting the figures and tables can be provided to qualified researchers upon

reasonable request and approval by the institutional Ethics Committee. Researchers who

meet the criteria for access to confidential data can request it through the Ethics Committee

of the Instituto Nacional de Ciencias Médicas y Nutrición Salvador Zubirán. Data requests

can be sent to the corresponding author at adrian.sotom@incmnsz.mx or to the Biobanco

Mexicano de Enfermedades Metabólicas at biomem@incmnsz.mx, who will coordinate

review with the institutional review board. The institutional research ethics committee can

be contacted at comite.etica.investigacion@incmnsz.mx. Page 1 and 2.

Minor comments

Statistical reporting

Please specify whether residual diagnostics were examined for the linear mixed-effects

models.

Clarify whether time was modeled as a continuous or categorical variable.

Response: We confirm that residual diagnostics were conducted for both models. Visual

inspection of histograms and standard Q-Q plots of the residuals supported the assumption

of approximate normality in both the leptin and adiponectin models. Time was modeled as a

continuous variable (in minutes), allowing the model to estimate a single average rate of

change in adipokine concentration per minute across the postprandial period. Both of these

details were already present in the Statistical Analysis section; however, we have reorganized

and expanded the relevant sentences to make them more explicit and easier to locate.

Change made: Added to the Statistical Analysis section: "In the linear mixed-effects models,

time was modeled as a continuous variable (in minutes), allowing the model to estimate a

single average rate of change in adipokine concentration per minute across the postprandial

period. Model assumptions were evaluated through residual diagnostics, including visual

inspection of histograms and standard Q-Q plots of the residuals, which supported the

assumption of approximate normality." Page 9

Terminology

In several places, “stable” is used without an explicit statistical reference. Consider replacing

with “did not change significantly” for precision.

Response: We have revised the manuscript to replace this term with “did not change

significantly” where appropriate to improve clarity and precision.

Figures

Figures are clear and informative. However, the color gradient representing BMI may be

difficult to interpret for readers with color vision deficiencies. Consider whether an

alternative or simplified representation is feasible.

Response: We have replaced the original yellow-to-red gradient for BMI with the viridis

color scale (option D), which is perceptually uniform and designed to remain interpretable

across the most common forms of color vision deficiency. Additionally, the green/red color

pair previously used to distinguish obesity groups has been replaced with the Okabe-Ito

palette (blue #0072B2 for non-obese; orange/amber #E69F00 for obese), which is widely

recommended for scientific figures requiring color vision accessibility (Okabe & Ito, 2002;

Wong, Nature Methods, 2011). The revised figures have been re-exported as TIFF files at

300 DPI with LZW compression, meeting PLOS ONE technical requirements.

Change made: All four figures (1A–1D) were regenerated using the viridis color scale for

BMI trajectories and the Okabe-Ito palette for obesity group comparisons.

Typographical and stylistic issues

Minor grammatical inconsistencies remain (e.g., spacing, punctuation around units). A final

careful proofreading is recommended.

Response: The manuscript has been carefully proofread, and minor typographical and

stylistic inconsistencies (including spacing and punctuation around units) have been

corrected throughout the text.

7. PLOS authors have the option to publish the peer review history of their article (what does

this mean?). If published, this will include your full peer review and any attached files. Yes

---

## [Editor Report · Decision Letter 2]

27 Apr 2026

PONE-D-25-57507R2Postprandial response of leptin and adiponectin to standardized high-carbohydrate and high-fat meals in adults: a cross-sectional studyPLOS One

Dear Dr. Soto-Mota,

Thank you for submitting your manuscript to PLOS ONE. I would like to congratulate the authors for their efforts in addressing the reviewers’ comments. I have some final considerations before submitting my recommendation to accept this manuscript. Therefore, I would like invite you to submit a revised version of the manuscript that addresses the points raised during the review process.

We look forward to receiving your revised manuscript.

Kind regards,

Neftali Eduardo Antonio-Villa, MD PhD

Academic Editor

PLOS One

Journal Requirements:

Additional Editor Comments :

•    The manuscript title describes the study as “cross-sectional,” whereas the cover letter refers to it as a “crossover study.” Please harmonize this accordingly.

•    Please carefully revise Tables 2 and 3 before publication. The reference level for meal type appears inconsistent between the Results text and the table footnotes. The Results state that women and the high-carbohydrate meal were used as reference categories, whereas the table footnotes state that “High in Fats” is the reference level.

•    In the Results, there appear to be some typos in the line: “relatively stablenon-significantly changing adiponectin concentrations.” This should be corrected to “non-significantly changing adiponectin concentrations” or “relatively stable adiponectin concentrations.”

•    The sentence stating that reductions in leptin “could attenuate satiety signaling and contribute to excessive energy intake” should be slightly softened, since satiety, appetite, and subsequent energy intake were not directly measured. I would suggest wording such as: “could potentially influence satiety-related pathways, although this hypothesis requires confirmation in studies directly assessing appetite and energy intake.”

•    Please revise minor typographical issues throughout the text, including “adipoectin,” spacing around references and punctuation, and inconsistent capitalization of “Sex,” “BMI,” and “Meal Type” when they are not table labels.

---

## [Author Response · Author response to Decision Letter 3]

27 Apr 2026

April 27th, 2026

Dear Dr. Antonio-Villa,

Thank you for the opportunity to revise our manuscript once more and for the constructive editorial comments. We have carefully addressed each point raised in your decision letter. All changes made in the revised manuscript are indicated with tracked changes, and the corresponding locations are referenced in our responses below. We hope that the revised manuscript is now suitable for publication in PLOS ONE. Please let us know if any additional information is required.

Additional Editor Comments

1. Study design terminology

Editor comment: The manuscript title describes the study as “cross-sectional,” whereas the cover letter refers to it as a “crossover study.” Please harmonize this accordingly.

Response: We thank the Editor for noting this inconsistency. The study design is cross-sectional with repeated postprandial measurements under two standardized meal conditions; it is not a crossover trial. The manuscript consistently and correctly describes the design as “cross-sectional” throughout the text, Methods, and in alignment with the STROBE reporting guideline. The error was in the previous cover letter, which has now been corrected. The revised cover letter accompanying this resubmission uses “cross-sectional study” to match the manuscript.

2. Reference level inconsistency between Results text and Tables 2 and 3

Editor comment: Please carefully revise Tables 2 and 3 before publication. The reference level for meal type appears inconsistent between the Results text and the table footnotes. The Results state that women and the high-carbohydrate meal were used as reference categories, whereas the table footnotes state that “High in Fats” is the reference level.

Response: We thank the Editor for drawing our attention to these issues. After careful review, we confirmed that the table footnotes are correct: the reference level for meal type in the mixed-effects models is the high-fat meal (i.e., “High in Fats”), and the reference for sex is Female. The error was in the Results text, which incorrectly stated that the high-carbohydrate meal was the reference category. We have corrected the Results text to read: “Women and the high-fat meal were used as reference categories for sex and meal type, respectively.” Additionally, we corrected the subsequent interpretation of the β coefficients. The coefficient for “Meal High-carbohydrate” (β = 5.93) now correctly refers to the high-carbohydrate meal (relative to the high-fat reference), and the interaction term (BMI × Meal High-carbohydrate) is described accordingly. The sentence about BMI-related increases being “weaker after the high-fat meal” was corrected to “weaker after the high-carbohydrate meal in comparison with the high-fat meal.”

3. Typographical errors in the Results section

Editor comment: In the Results, there appear to be some typos in the line: “relatively stablenon-significantly changing adiponectin concentrations.” This should be corrected to “non-significantly changing adiponectin concentrations” or “relatively stable adiponectin concentrations.”

Response: Thank you for bringing this up. The phrase has been corrected to “relatively stable adiponectin concentrations,” which we believe is the clearest phrasing.

4. Softening the leptin–satiety–energy intake claim

Editor comment: The sentence stating that reductions in leptin “could attenuate satiety signaling and contribute to excessive energy intake” should be slightly softened, since satiety, appetite, and subsequent energy intake were not directly measured. I would suggest wording such as: “could potentially influence satiety-related pathways, although this hypothesis requires confirmation in studies directly assessing appetite and energy intake.”

Response: Thank you for this suggestion. We have adopted it in both the Abstract (Conclusions) and the Discussion.

5. Minor typographical issues

Editor comment: Please revise minor typographical issues throughout the text, including “adipoectin,” spacing around references and punctuation, and inconsistent capitalization of “Sex,” “BMI,” and “Meal Type” when they are not table labels.

Response: Thank you. We corrected the typo “adipoectin” → “adiponectin” in the Results section (adiponectin model description). We have also reviewed the manuscript for spacing around references, punctuation, and capitalization consistency. In the running text, “sex,” “BMI,” and “meal type” are now used in lowercase except when they appear as table row labels.

6. Reference list review

Journal requirement: Please review your reference list to ensure that it is complete and correct. If you have cited papers that have been retracted, please include the rationale for doing so in the manuscript text, or remove these references and replace them with relevant current references.

Response: We have reviewed the reference list and confirmed that all 22 cited works are current, accessible, and none have been retracted. No changes to the reference list were necessary.

Thank you again for your consideration.

---

## [Editor Report · Decision Letter 3]

30 Apr 2026

Postprandial response of leptin and adiponectin to standardized high-carbohydrate and high-fat meals in adults: a cross-sectional study

PONE-D-25-57507R3

Dear Dr. Soto-Mota,

We’re pleased to inform you that your manuscript has been judged scientifically suitable for publication and will be formally accepted for publication once it meets all outstanding technical requirements.

Kind regards,

Neftali Eduardo Antonio-Villa, MD PhD

Academic Editor

PLOS One

Additional Editor Comments (optional):

Congratulations for the exceptional work!
---

## [Editor Report · Acceptance letter]

PONE-D-25-57507R3

PLOS One

Dear Dr. Soto-Mota,

I'm pleased to inform you that your manuscript has been deemed suitable for publication in PLOS One. Congratulations! Your manuscript is now being handed over to our production team.

Kind regards,

on behalf of

Dr. Neftali Eduardo Antonio-Villa

Academic Editor

PLOS One